materials science

5-*N*-bis(amidopropyltriethoxysilyl) nicotinic acid (ANA-Si), core–shell–shell nanostructured composite, photoluminescence, silica shell, low-temperature phosphorescence

**Author for correspondence:**
Wenxian Li
e-mail: nmglwx@163.com

This article has been edited by the Royal Society of Chemistry, including the commissioning, peer review process and editorial aspects up to the point of acceptance.

# Synthesis and photoluminescence properties of silica-modified SiO₂@ANA-Si-Tb@SiO₂, SiO₂@ANA-Si-Tb-L@SiO₂ core–shell–shell nanostructured composites

Lina Feng[1], Wenxian Li[1], Jinrong Bao[1], Yushan Zheng[2], Yilian Li[1], Yangyang Ma[1], Kuisuo Yang[1], Yan Qiao[1] and Anping Wu[1]

[1]Inner Mongolia Key Laboratory of Chemistry and Physics of Rare Earth Materials, School of Chemistry and Chemical Engineering, Inner Mongolia University, Hohhot 010021, People's Republic of China
[2]Inner Mongolia Autonomous Region Food Inspection Test Center, Hohhot 010010, People's Republic of China

(iD) W-XL, 0000-0002-0938-6910

Three novel core–shell nanostructured composites SiO₂@ANA-Si-Tb, SiO₂@ANA-Si-Tb-L (L = second ligand) with SiO₂ as the core and terbium organic complex as the shell were successfully synthesized. The core and shell were connected together by covalent bonds. The terbium ion was coordinated with organic ligand-forming terbium organic complex in the shell layer. The organosilane (HOOCC₅H₄NN(CONH(CH₂)₃Si(OCH₂CH₃)₃)₂ (abbreviated as ANA-Si) was used as the first ligand and 1,10-phenanthroline (phen) or 2-thenoyltrifluoroacetone (TTA) was used as the second ligand. Furthermore, silica-modified SiO₂@ANA-Si-Tb@SiO₂, SiO₂@ANA-Si-Tb-L@SiO₂ core–shell–shell nanostructured composites were also synthesized by sol–gel chemical route, which involved the hydrolysis and polycondensation processes of tetraethoxysilane (TEOS) using cetyltrimethyl ammonium bromide (CTAB) as a surface-active agent. An amorphous silica shell was coated around the SiO₂@ANA-Si-Tb, SiO₂@ANA-Si-Tb-L core–shell nanostructured composites. The core–shell and core–shell–shell nanostructured composites exhibited excellent luminescence in the solid state.

Meanwhile, an improved luminescent stability property of the core–shell–shell nanostructured composites was observed for the aqueous solution. This type of core–shell–shell nanostructured composites exhibited bright luminescence, high stability and good solubility, which may present potential applications in the fields of optoelectronic devices, bio-imaging, medical diagnosis and study on the structure of function composite materials.

## 1. Introduction

Over recent years, core–shell nanostructured composite materials, which connect different functional components integrated into one unit, have attracted increased attention owing to their interesting properties and broad range of applications in catalysis [1–6], bio-nanotechnology [7–11], materials chemistry [12], optical devices [13–19], electronics [20–23] and magnetic devices [24–28]. Among the number of core–shell nanostructured composite materials [29–32], coating desirable compounds onto a core to form luminescence materials with the required photophysical properties for particular applications have emerged as one of the research hotspots in recent years. Compared with unary substance, this core–shell nanostructured composite material often exhibits improved physical and chemical properties [33–36], such as stability, multi-functionality, luminous intensity, fluorescence quantum efficiency and fluorescence lifetime.

In core–shell nanostructured composite materials, silica particles are very important core material because of their aqueous solubility, surface tailorability, low cytotoxicity and low cost [37]. Moreover, the surface of $SiO_2$ has many active hydroxyl groups and can be chemically bonded to a substance with a functional property. These core–shell nanostructured composites not only keep core materials stable but also have shell layer-specific physico-chemical properties. So far, there are two commonly used methods to fabricate core–shell nanostructured composites. One method is direct precipitation, in which compounds can be deposited on the surface of the $SiO_2$ core and form core–shell nanostructured composites. In most cases, however, the degree of surface coverage is low and the coating is not uniform. Meanwhile, the prepared composite is unstable, and the core–shell nanostructured composite may easily collapse. Another method is silane coupling agent method. Here, the silane coupling agent is a bifunctional organics (denoted as $Y(CH_2)_nSiX_3$). Y expresses the organic functional groups, such as amino, carboxyl and double nitrogen, where the Y group could coordinate to rare earth ions forming rare earth organic complexes. X expresses the alkoxy. The Si–O–Si chemical bonds are constructed after hydrolysis and polycondensation processes of X groups and the hydroxyl of $SiO_2$ surface. In such a method, the silane coupling agent connects $SiO_2$ spheres and rare earth complexes together. The core–shell nanostructured composites are stable and the thickness of the shell is easily controlled. Therefore, the silane coupling agent method is considered as an effective and popular strategy.

Nowadays, there is sufficient research on core–shell nanostructured composite materials with $SiO_2$ as core and inorganic material as cladding layer [38–41]. However, the rare earth organic complexes as cladding layer have not been extensively involved. In order to obtain excellent luminescence functional materials, we chose to use the rare earth organic complexes as a coating layer, which can make full use of the 'antenna effect' of organic ligands achieving higher luminescent efficiency [42]. In our prior work, we have successfully studied the change of luminescence properties forming core–shell nanostructured composites. In addition, as the thickness of rare earth organic complex coating layer is nanometre magnitude, it can significantly save rare earth resources and greatly reduce the production cost [43,44]. To the best of our study, although they exhibited excellent luminescence in the solid state, they usually presented poor luminescent stability properties under aqueous medium. The rare earth organic complex shell was easily quenched in the aqueous environment, which restricted the practical application of them. Therefore, it is significantly necessary to obtain the silica-modified core–shell–shell nanostructured composites and improve luminescent stability properties. The silica shell plays an important role in preventing rare earth core–shell nanostructured composites from quenching of external environment and also improving the solubility and luminescent stability of the core–shell nanostructured composite material. For example, Ansari prepared hierarchical $CePO_4$:Tb@$LaPO_4$@$SiO_2$ core–shell–shell nanostructured composite that had significant application to enhance the solubility, colloidal stability character and high luminescence properties [45]. Therefore, the silica-modified sol–gel technique is an ideal option to improve luminescent stability and the solubility of rare earth core–shell nanostructured composites. In addition, such silica-modified core–shell–shell nanostructured composites simultaneously show excellent properties with regard to non-toxicity and luminescence

**Figure 1.** The synthesis route of ANA-Si.

that plays a significant role in the development of bio-imaging, medical diagnosis and biological labelling [46,47].

In this report, a bifunctional silane coupling agent method was presented for the synthesis of SiO$_2$@ANA-Si-Tb and SiO$_2$@ANA-Si-Tb-L core–shell nanostructured composites. Specifically, the synthetic approach involved the preparation of organosilane ANA-Si, which acted as a 'functional bridge molecular'. The silica core and the terbium organic complex shell were connected together by a hydrolysation process forming Si–O–Si covalent bonds. Furthermore, the silica-modified SiO$_2$@ANA-Si-Tb@SiO$_2$ and SiO$_2$@ANA-Si-Tb-L@SiO$_2$ core–shell–shell nanostructured composites were prepared using the TEOS-CTAB sol–gel chemical route. In such a strategy, an amorphous silica shell was successfully coated on the surface of SiO$_2$@ANA-Si-Tb and SiO$_2$@ANA-Si-Tb-L using CTAB as a surface-active agent. In contrast to the corresponding core–shell nanostructured composites, the SiO$_2$@ANA-Si-Tb@SiO$_2$ and SiO$_2$@ANA-Si-Tb-L@SiO$_2$ core–shell–shell nanostructured composites showed improved photoluminescence properties and luminescent stability in aqueous solution. Therefore, the silica-modified SiO$_2$@ANA-Si-Tb@SiO$_2$ and SiO$_2$@ANA-Si-Tb-L@SiO$_2$ core–shell–shell nanostructured composites were extending their potential application in photonics-based biomedical sciences.

# 2. Experimental section

## 2.1. Chemicals and reagents

Tb$_4$O$_7$ (99.99%), ammonia (25–28%), urea (99%) and cetyltrimethyl ammonium bromide (CTAB, 99%) were purchased from Sigma-Aldrich (Steinheim, Germany). 3-(triethoxysilyl)-Propyl isocyanate (TEPIC, 95%), 5-aminonicotinic acid (98%), 1,10-phenanthroline (phen, 99%), 2-thenoyltrifluoroacetone (TTA, 98%), tetraethoxysilane (TEOS, greater than 99%) were purchased from Sinopharm Chemical Reagent Co., Ltd (Shanghai, China). All other chemicals were of analytical grade and used as received without further purification. The terbium perchlorate (Tb(ClO$_4$)$_3 \cdot n$H$_2$O) was prepared by dissolving Tb$_4$O$_7$ (99.99%) in HClO$_4$ (1 mol l$^{-1}$) and then evaporated and dried in vacuum.

## 2.2. Synthesis of organosilane (ANA-Si)

5-Aminonicotinic acid (0.28 g) and pyridine (25.0 ml) were added to a 100 ml three-necked round-bottom flask and refluxed for 2 h under magnetic stirring. Then, 4 mmol (1.0 ml) TEPIC was added dropwise. After the reaction mixture had been heated for about 12 h, unreacted pyridine was distilled off under reduced pressure. The solid residue was further washed with absolute ether and then dried under vacuum at 50°C to obtain ANA-Si as a white powder. The synthesis route is shown in figure 1. Yield: 75%. Anal. Calcd. of C$_{26}$H$_{48}$N$_4$O$_{10}$Si$_2$: C, 49.35%; H, 7.65%; N, 8.85%; Found: C, 48.97%; H, 7.17%; N, 9.11%. $^1$H NMR (CDCl$_3$): $\delta$ 0.56 ppm (4H), $\delta$ 1.12–1.56 ppm (22H), $\delta$ 3.57–3.72 ppm (12H), $\delta$ 5.95 ppm (2H), $\delta$ 8.63–7.87 ppm (3H) and $\delta$ 13.04 ppm (1H).

## 2.3. Synthesis of silica cores

The highly monodisperse silica spheres were synthesized by the well-known Stöber process [48]. Typically, 1.7 ml of TEOS was rapidly dropped into a mixture that consisted of 40 ml anhydrous ethanol, 4 ml deionized water and 1.7 ml ammonia in a 100 ml single-neck round-bottom flask under vigorous stirring. The reaction mixture was then placed in an automatic microwave synthesizer. (The pressure was 15 PSI, the power was 150 W, the rotational speed was medium speed and the reaction temperature was 50°C.) After 4 h reaction time, the resulting white silica suspension was centrifugally separated and washed with ethanol and deionized water several times after drying at 50°C in an oven.

## 2.4. Synthesis of SiO$_2$@ANA-Si

For the preparation of SiO$_2$@ANA-Si spheres, briefly, 0.10 g of SiO$_2$ spheres were ultrasonically dispersed into a 15 ml mixed solvent of ethanol (10 ml) and deionized water (5 ml) forming a homogeneous dispersion, followed by dropwise addition of anhydrous ethanol (15 ml) containing ANA-Si (0.20 g). Then, the mixture solution was stirred at room temperature for 36 h to obtain a white homogeneous mixture. Afterward, the products were centrifuged, washed thoroughly with anhydrous ethanol successively and repeatedly and dried in an oven overnight at 50°C.

## 2.5. Synthesis of SiO$_2$@ANA-Si-Tb and SiO$_2$@ANA-Si-Tb-L

Typically, 0.15 g SiO$_2$@ANA-Si spheres were dispersed in 15 ml anhydrous ethanol through ultrasonication. Afterward, 0.07 g of Tb(ClO$_4$)$_3$·$n$H$_2$O was dissolved in 5 ml anhydrous ethanol and was added dropwise into the above solutions under vigorous stirring. The reaction mixture was heated at 50°C for 5 h forming a white precipitate. Then, the resulting white precipitate was collected after centrifugation and dried in an oven overnight at 50°C. The synthetic procedure for the synthesis of SiO$_2$@ANA-Si-Tb-L was as follows: 0.15 g SiO$_2$@ANA-Si spheres and 0.14 g second ligand L (phen or TTA) were dispersed in 15 ml anhydrous ethanol. Then, 0.20 g Tb(ClO$_4$)$_3$·$n$H$_2$O dissolved in 5 ml anhydrous ethanol was added, and the mixture was heated at 50°C for 5 h. The solution colour changed from white to light pink. Finally, the resulting precipitate was collected by centrifugation and dried in an oven overnight at 50°C.

## 2.6. Synthesis of SiO$_2$@ANA-Si-Tb@SiO$_2$ and SiO$_2$@ANA-Si-Tb-L@SiO$_2$

The Stöber sol–gel technique was popular for silica surface modification. The specific procedure was as follows: the core–shell nanostructured composites SiO$_2$@ANA-Si-Tb or SiO$_2$@ANA-Si-Tb-L (0.10 g), CTAB (0.12 g) and urea (0.10 g) were placed in a 50 ml bottle and dissolved in 18-ml mixed solvent of ethanol (15 ml) and deionized water (3 ml). After stirring the mixture for 10 min at room temperature, 0.70 ml TEOS dissolved in 5 ml anhydrous ethanol was added dropwise to the above dispersion to get a white suspension. The solution mixture was continuously stirred for 24 h, and the final precipitate was separated using centrifugation, washed with ethanol and dried at 50°C.

## 2.7. Apparatus and instrumentation

[1]H NMR spectroscopy was performed on a 600 M liquid [1]H NMR instrument (Bruker AVANCE III) using CDCl$_3$ as the solvent and dioxane as the internal standard. Elemental analyses were performed with an elemental C, H, N analyser. The structure and surface morphology of microspheres were studied with a scanning electron microscope (SEM, Hitachi S-4800, Japan) and transmission electron microscope (TEM, FEI Tecnai F20, USA) accompanied by energy-dispersive X-ray spectroscopy (EDX) to examine the chemical composition. Fourier transform infrared spectra (FT-IR) were measured on a Bruker VERTEX70 spectrometer at 500–4000 cm$^{-1}$ using the KBr pellet method. X-ray powder diffraction (XRD, RIGAKU, Japan) was measured using a 21 kW extra power X-ray diffractometer using Cu K$\alpha$ radiation ($\lambda = 1.5405$ Å) over an angle range from 5° to 80°. The photoluminescence spectra, quantum yields and lifetimes of the solid powder samples were determined using a fluorescence spectrometer at room temperature (FL, Edinburgh Instruments FLS098, UK). The phosphorescence spectra of the solid powder samples were monitored using an FLS980 spectrometer at 77 K.

## 2.8. Test methodology for the quantum yields

The absolute fluorescence quantum yield of the solid powder sample was measured by an integrating sphere using the photoluminescence spectrometer at room temperature. First, the excitation and emission spectra of the sample should be acquired in order to identify the best excitation wavelength and emission range. Subsequently, the integrating sphere was put in the sample chamber. In powder samples, the blank sample measurements were made by replacing the PTFE powder vessel with the BENFLEC scattering platform. An emission spectrum covering the excitation light and the full-emission spectrum was recorded. This measurement should be done with 0.1 nm step size and 0.3 s integration time, 1–5 repeats (depending on the accuracy required). The counts per second on the emission detector should be monitored until there was sufficient intensity on the detector (500 000–1 million counts s$^{-1}$).

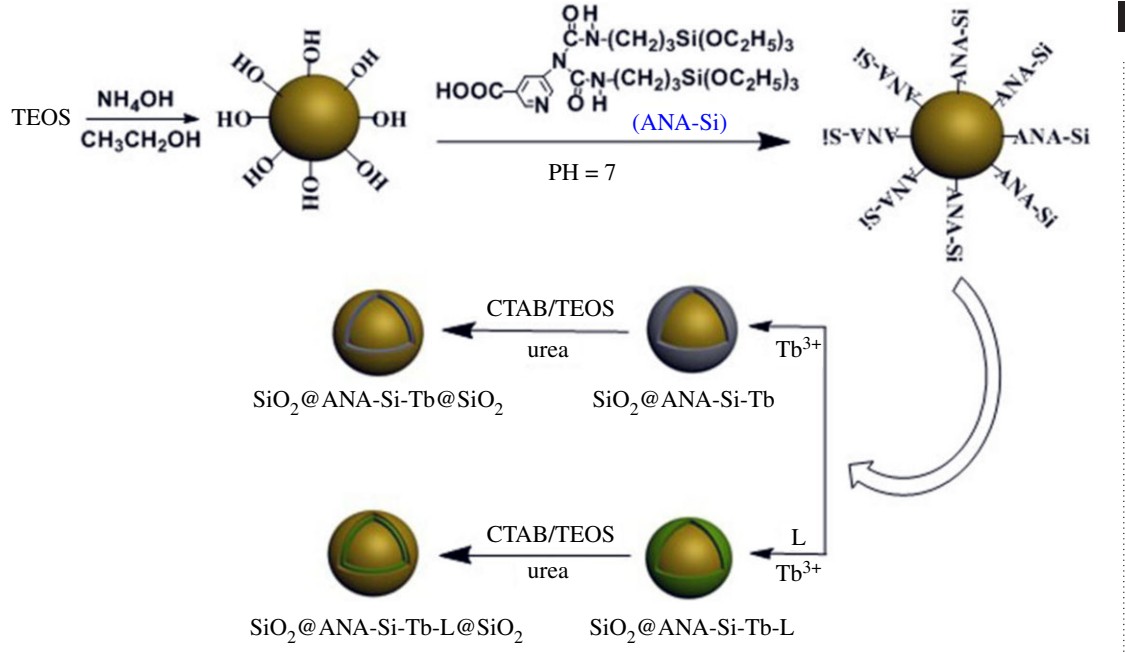

**Scheme 1.** Schematic illustration of formation mechanism for core–shell and core–shell–shell nanostructured composites.

Then, the emission spectrum of the solid powder samples covering the excitation light and the full-emission spectrum were recorded. This measurement should be done with the same settings as before. Finally, the quantum yield function was used to select the spectra and define the calculation region for scatter and emission. After this was completed, the quantum yield was obtained.

# 3. Results and discussion

## 3.1. The formation mechanism of $SiO_2$@ANA-Si-Tb, $SiO_2$@ANA-Si-Tb-L and $SiO_2$@ANA-Si-Tb@SiO_2$, $SiO_2$@ANA-Si-Tb-L@SiO_2$

The synthesis strategy for fabricating core–shell and core–shell–shell nanostructured composites is presented in scheme 1. In the first step, the synthetic approach involved the preparation of silica spheres and bifunctional organosilane ANA-Si. The organosilane acted as a 'functional bridge molecular', connecting the silica core and the terbium organic complex shell together by a hydrolysation forming Si–O–Si covalent bonds. In this process, organosilane ANA-Si was successfully grafted onto the surface of silica spheres. Then, the carboxyl group of ANA-Si could coordinate to terbium ions. Additionally, the introduction of the second ligand (phen and TTA) could also coordinate to terbium ions by double nitrogen atoms or oxygen atom and sensitize the luminescence of terbium ions. The core–shell nanostructured composites were obtained. Second, the silica-modified core–shell–shell nanostructured composites were synthesized. An amorphous silica shell was uniformly coated around these as-prepared core–shell nanostructured composites by the sol–gel chemical route. The hydrolysis and polycondensation process of TEOS was involved using CTAB as a surface-active agent. The silica-modified core–shell–shell nanostructured composites not only protected the core–shell nanostructured composites from the surrounding environment but also improved luminescent stability of the core–shell–shell nanostructured composites. Moreover, the surface Si–OH groups played a significant role in the functionalization and their affinity with biomacromolecules, which was used in biological systems for the detection of various analytes.

## 3.2. Morphology and structure

### 3.2.1. The TEM of the $SiO_2$@ANA-Si-Tb and $SiO_2$@ANA-Si-Tb-L

The morphology and size of the $SiO_2$ are shown in figure 2*a,b*. Based on the SEM and TEM micrographs, $SiO_2$ had a spherical shape and smooth surface with an average diameter of 181 nm. Figure 2*c,d* shows

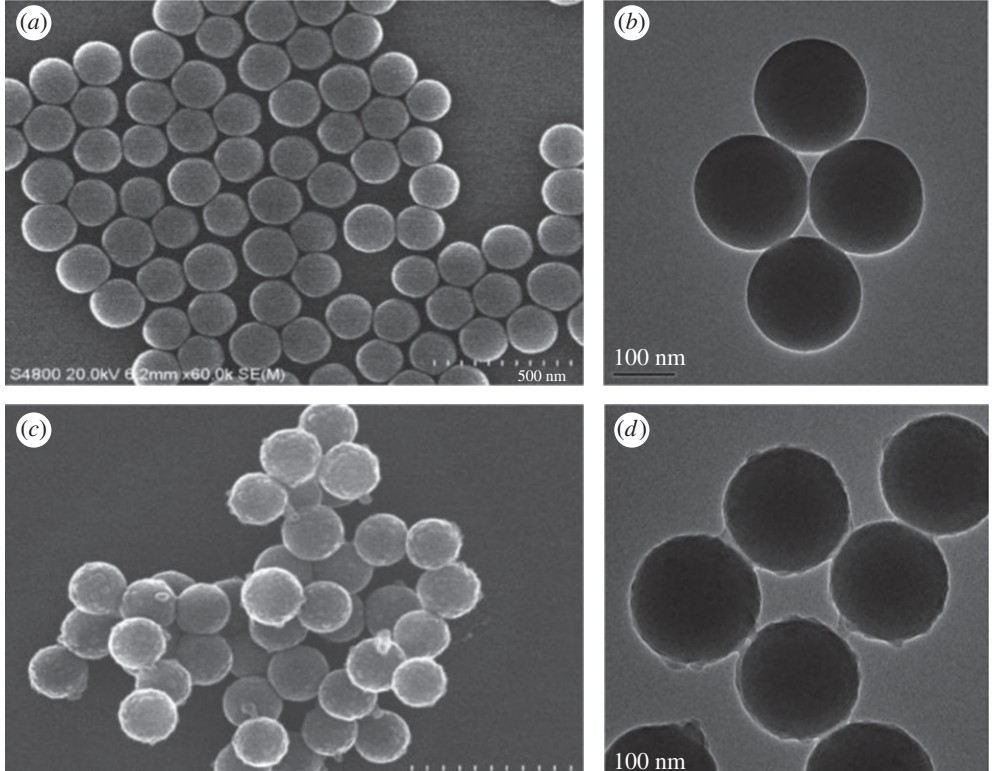

**Figure 2.** SEM and TEM images of SiO$_2$ (a,b) and SiO$_2$@ANA-Si (c,d).

the SEM and TEM images of SiO$_2$@ANA-Si particles. As can be seen from a low-magnification SEM image, these SiO$_2$@ANA-Si particles exhibited perfect uniformity and monodispersity. TEM image revealed that they possessed rough surfaces with a diameter of 188 nm. Simultaneously, the formation of SiO$_2$@ANA-Si-Tb, SiO$_2$@ANA-Si-Tb-phen and SiO$_2$@ANA-Si-Tb-TTA core–shell nanostructured composites was clearly verified by TEM. As shown in figure 3a–c, these core–shell nanostructured composites were still spherical and non-aggregated but slightly larger than the SiO$_2$@ANA-Si particles. The changes in diameter from 181 nm for SiO$_2$ sphere to 193 nm for SiO$_2$@ANA-Si-Tb and SiO$_2$@ANA-Si-Tb-L indicated that the terbium organic complexes were successfully coated on the surface of SiO$_2$ spheres with a thickness of about 6 nm. In order to study the elemental composition of the core–shell nanostructured composites, the as-prepared SiO$_2$@ANA-Si-Tb, SiO$_2$@ANA-Si-Tb-phen and SiO$_2$@ANA-Si-Tb-TTA microspheres were subsequently analysed by EDX spectrometer (figure 3d and electronic supplementary material, figure S1a). The presence of Si, O, N, Cl and Tb atoms in the EDX spectra suggested the formation of the SiO$_2$@ANA-Si-Tb, SiO$_2$@ANA-Si-Tb-phen and SiO$_2$@ANA-Si-Tb-TTA core–shell nanostructured composites. Furthermore, the element mappings of Si, O and Tb in SiO$_2$@ANA-Si-Tb composite (figure 4a–d) revealed that O and Tb were uniformly distributed in the silica-based sphere-like area.

### 3.2.2. The TEM of the SiO$_2$@ANA-Si-Tb@SiO$_2$ and SiO$_2$@ANA-Si-Tb-L@SiO$_2$

In order to extend the application of core–shell nanostructured composite material in the biomedical field, the silica-modified SiO$_2$@ANA-Si-Tb@SiO$_2$, SiO$_2$@ANA-Si-Tb-phen@SiO$_2$ and SiO$_2$@ANA-Si-Tb-TTA@SiO$_2$ core–shell–shell nanostructured composites were also synthesized. Further information about the shell growth was provided using TEM analysis. Figure 5 shows the TEM micrograph of SiO$_2$@ANA-Si-Tb@SiO$_2$, SiO$_2$@ANA-Si-Tb-phen@SiO$_2$ and SiO$_2$@ANA-Si-Tb-TTA@SiO$_2$ core–shell–shell nanostructured composites. There was a clear morphological difference between the core–shell nanostructured composites and the silica-modified core–shell–shell nanostructured composites; wherein, a rougher SiO$_2$ layer about 8–12 nm thickness was coated around the surface of SiO$_2$@ANA-Si-Tb, SiO$_2$@ANA-Si-Tb-phen and SiO$_2$@ANA-Si-Tb-TTA core–shell nanostructured composites. Moreover, the EDX spectrometer was employed to the core–shell–shell nanostructured composites and illustrated that there still existed the Si, O, N, Cl and Tb atoms in figure 5d and electronic

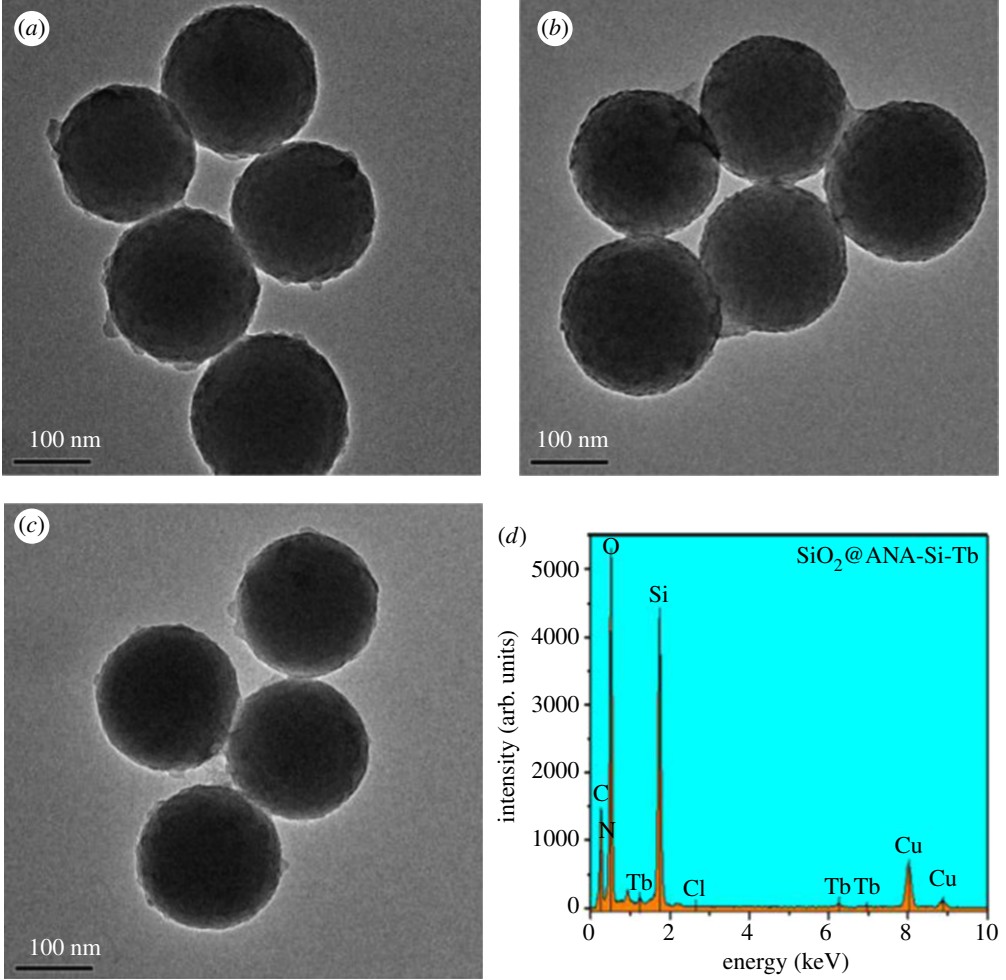

**Figure 3.** TEM images of SiO$_2$@ANA-Si-Tb (*a*), SiO$_2$@ANA-Si-Tb-phen (*b*), SiO$_2$@ANA-Si-Tb-TTA (*c*) and EDX spectrum of SiO$_2$@ANA-Si-Tb (*d*).

supplementary material, figure S1b; however, the Tb content decreased from 2.32% for SiO$_2$@ANA-Si-Tb (figure 3*d*) to 0.55% for SiO$_2$@ANA-Si-Tb@SiO$_2$ (figure 5*d*). Owing to an amorphous silica layer coated onto the surface of SiO$_2$@ANA-Si-Tb, SiO$_2$@ANA-Si-Tb-phen and SiO$_2$@ANA-Si-Tb-TTA core–shell nanostructured composites, the total quality of core–shell–shell nanostructured composites was increased, eventually resulting in the reduction of Tb content compared with corresponding core–shell nanostructured composites.

## 3.3. Infrared spectra

### 3.3.1. The FT-IR spectra of the SiO$_2$@ANA-Si-Tb and SiO$_2$@ANA-Si-Tb@SiO$_2$

Further to verify the purity and chemical compositions of core–shell and core–shell–shell nanostructured composites, FT-IR spectra of ANA-Si, SiO$_2$, SiO$_2$@ANA-Si, SiO$_2$@ANA-Si-Tb and SiO$_2$@ANA-Si-Tb@SiO$_2$ are displayed in figure 6*a*–*e*. The appearance of the characteristic bands ANA-Si (figure 6*a*) at 1640 and 1556 cm$^{-1}$ was attributed to the stretching vibration of –CONH– bonds, indicating that ANA-Si had been successfully synthesized by the amidation reaction with 5-aminonicotinic acid and 3-(triethoxysilyl)-propyl isocyanate. The stretching vibration of –C=O– (COOH) appeared at 1701 cm$^{-1}$. The characteristic band of SiO$_2$ (figure 6*b*) at 1098 cm$^{-1}$ was attributed to the stretching vibration of Si–O–Si group, and Si–OH group was identified at 952 cm$^{-1}$. In the spectra of SiO$_2$@ANA-Si (figure 6*c*), the appearance of the characteristic band at 1701 cm$^{-1}$ was ascribed as the stretching vibration of –C=O– (COOH). The characteristic bands at 1644 and 1560 cm$^{-1}$ were originated from the stretching vibration of –CONH–, which further confirmed the reaction of 'functional bridge molecular' ANA-Si with the hydroxyl groups on the surface of SiO$_2$. FT-IR spectrum of SiO$_2$@ANA-Si-Tb (figure 6*d*) showed the

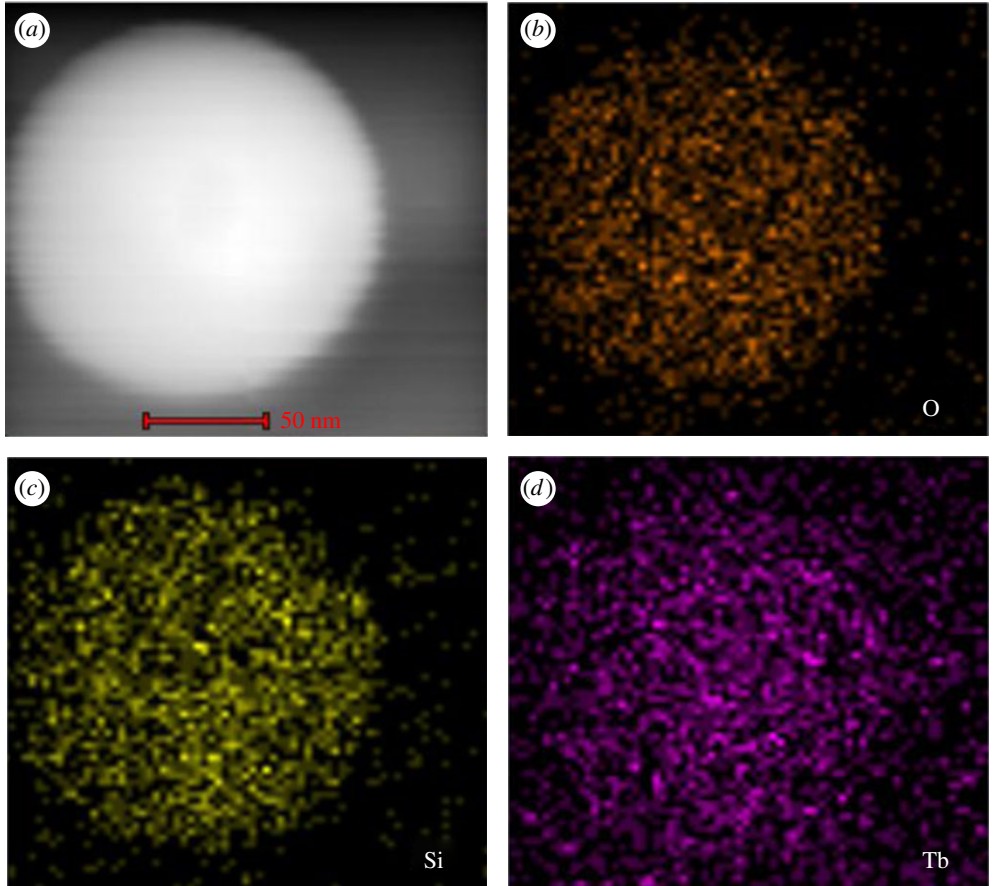

**Figure 4.** Bright-field scanning transmission electron microscopy (STEM) image of a single SiO₂@ANA-Si-Tb (*a*) and the associated EDX element mappings images (*b–d*).

characteristic bands of –C=O– (COOH) bonds at 1688 cm$^{-1}$. There was an obvious change in the absorption band compared with that of SiO₂@ANA-Si, indicating that Tb³⁺ ions coordinated with the oxygen atoms of carbonyl group. In addition, observed characteristic infrared bands at 1145, 1084 and 625 cm$^{-1}$ corresponded to the vibration of perchlorate group ($ClO_4^-$). As can be seen from the FT-IR spectrum of SiO₂@ANA-Si-Tb@SiO₂ (figure 6*e*), there was a significant change compared to the spectrum of SiO₂@ANA-Si-Tb (figure 6*d*). However, a strong band at 1089 cm$^{-1}$ was similar to that of the spectrum of SiO₂ (figure 6*b*), which could confirm that the SiO₂ shell had successfully coated onto the surface of SiO₂@ANA-Si-Tb core–shell nanostructured composites.

### 3.3.2. The FT-IR spectra of the SiO₂@ANA-Si-Tb-L and SiO₂@ANA-Si-Tb-L@SiO₂

After further encapsulation by the second ligand and SiO₂ shell, the formation process of core–shell and core–shell–shell nanostructured composites was also investigated by FT-IR spectra. Figure 7*a–c* shows the FT-IR spectra of phen, SiO₂@ANA-Si-Tb-phen and SiO₂@ANA-Si-Tb-phen@SiO₂. Comparing the spectra SiO₂@ANA-Si-Tb-phen (figure 7*b*) with SiO₂@ANA-Si (figure 6*c*), the stretching vibration of –C=O– (COOH) red-shifted to 1691 cm$^{-1}$, and the stretching vibration of –CONH– red-shifted to 1636 and 1519 cm$^{-1}$, implying that the –C=O– (COOH) of ANA-Si was coordinated with Tb³⁺. Moreover, the stretching vibration of C=N in the spectra of free phen (figure 7*a*) was located at about 1587 cm$^{-1}$, however, which had shifted to lower frequencies at 1560 cm$^{-1}$ in the spectra of the SiO₂@ANA-Si-Tb-phen (figure 7*b*), suggesting that the Tb³⁺ ion coordinated with double nitrogen atoms of phen. In the FT-IR spectrum of SiO₂@ANA-Si-Tb-phen@SiO₂ (figure 7*c*), there was a significant change compared to the spectra of SiO₂@ANA-Si-Tb-phen (figure 7*b*). However, a strong band at 1091 cm$^{-1}$ was similar to that of the spectrum of SiO₂ (figure 6*b*), indicating that the SiO₂ shell was coated on the surface of SiO₂@ANA-Si-Tb-phen.

Figure 7*d–f* shows the FT-IR spectra of TTA, SiO₂@ANA-Si-Tb-TTA and SiO₂@ANA-Si-Tb-TTA@SiO₂. Most important absorption band was similar to the one shown in figure 7*a–c*. In the spectra of

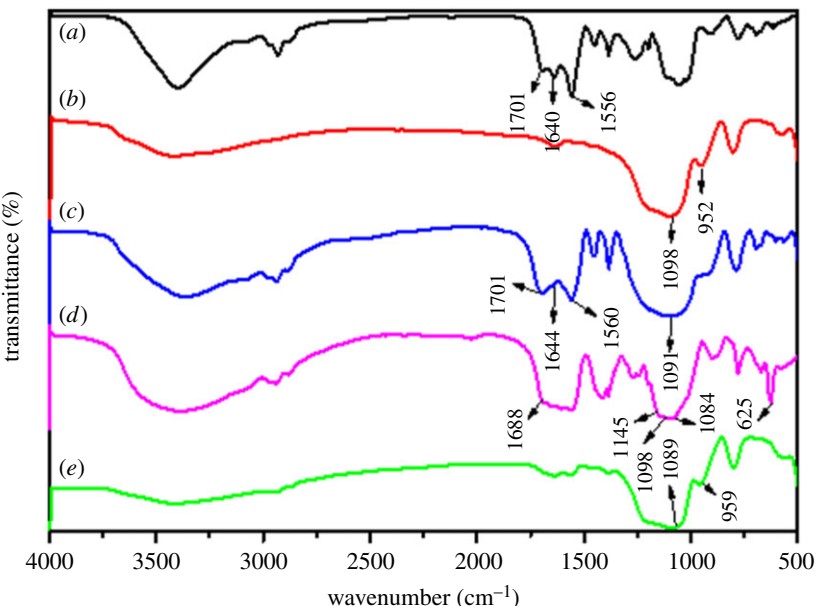

**Figure 5.** TEM images of SiO$_2$@ANA-Si-Tb@SiO$_2$ (*a*), SiO$_2$@ANA-Si-Tb-phen@SiO$_2$ (*b*), SiO$_2$@ANA-Si-Tb-TTA@SiO$_2$ (*c*) and EDX spectrum of SiO$_2$@ANA-Si-Tb@SiO$_2$ (*d*).

**Figure 6.** FT-IR spectra of ANA-Si (*a*), SiO$_2$ (*b*), SiO$_2$@ANA-Si (*c*), SiO$_2$@ANA-Si-Tb (*d*) and SiO$_2$@ANA-Si-Tb@SiO$_2$ (*e*).

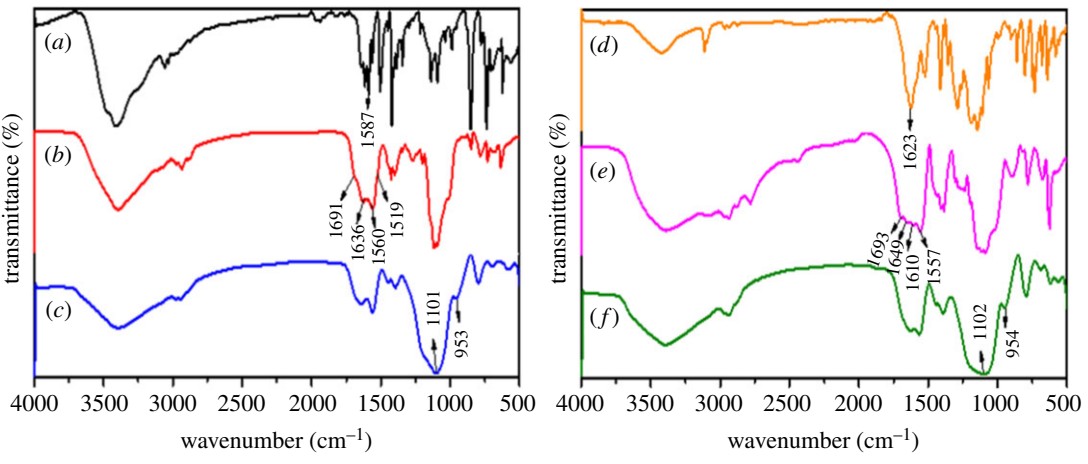

**Figure 7.** FT-IR spectra of phen (*a*), SiO$_2$@ANA-Si-Tb-phen (*b*), SiO$_2$@ANA-Si-Tb-phen@SiO$_2$ (*c*), TTA (*d*), SiO$_2$@ANA-Si-Tb-TTA (*e*) and SiO$_2$@ANA-Si-Tb-TTA@SiO$_2$ (*f*).

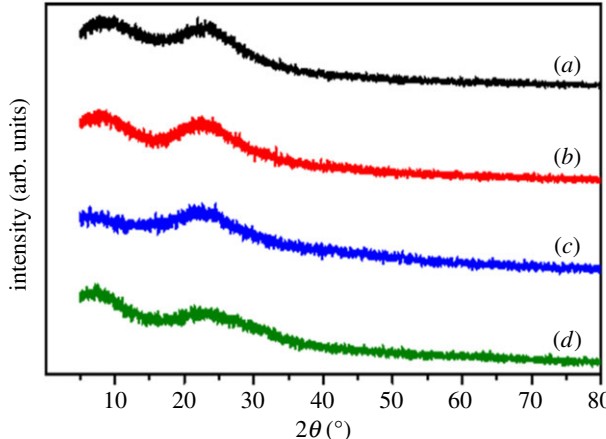

**Figure 8.** XRD pattern of SiO$_2$ (*a*), SiO$_2$@ANA-Si (*b*), SiO$_2$@ANA-Si-Tb (*c*) and SiO$_2$@ANA-Si-Tb@SiO$_2$ (*d*).

SiO$_2$@ANA-Si-Tb-TTA (figure 7*e*), the stretching vibration of –C=O– was shifted to lower wavenumber, from 1623 cm$^{-1}$ of free TTA (figure 7*d*) to about 1610 cm$^{-1}$, indicating that the Tb$^{3+}$ ions coordinated with carbonyl oxygen atom of TTA. Furthermore, in the FT-IR spectrum of SiO$_2$@ANA-Si-Tb-TTA@SiO$_2$ (figure 7*f*), silica-modified surface of SiO$_2$@ANA-Si-Tb-TTA was confirmed by the characteristic bands of SiO$_2$ at 1102 and 954 cm$^{-1}$.

## 3.4. XRD analysis

The composition and the structure of the solid powder samples were examined using XRD. Figure 8 shows the XRD patterns of SiO$_2$, SiO$_2$@ANA-Si, SiO$_2$@ANA-Si-Tb and SiO$_2$@ANA-Si-Tb@SiO$_2$, respectively. For SiO$_2$ directly formed from Stöber method (figure 8*a*), there were two broad bands centred at $2\theta = 7$–8° and 23°, which were identical with the standard XRD pattern for amorphous SiO$_2$. After surface modification with organosilane (ANA-Si), as shown in figure 8*b*, no diffraction peak was observed except for the broad bands, which were the characteristic peak for amorphous SiO$_2$. The XRD patterns of SiO$_2$@ANA-Si-Tb (figure 8*c*) and the intensity of the diffraction pattern appeared at $2\theta = 7$–8° became weaker. Careful viewing showed that there were some weak peaks centred at 6.4°, 7.9°, 9.1°, 10.2° and 11.1°, also providing evidence for the Tb-(ANA-Si)-ClO$_4$ complex coated onto the surface of SiO$_2$ core. As illustrated in figure 8*d*, the two broad bands of SiO$_2$@ANA-Si-Tb@SiO$_2$ core–shell–shell nanostructured composite appearing at $2\theta = 7$–8° and 23° were in coincidence with the standard XRD pattern for SiO$_2$ core (figure 8*a*), which further illustrated that an amorphous SiO$_2$ was coated onto the surface of SiO$_2$@ANA-Si-Tb.

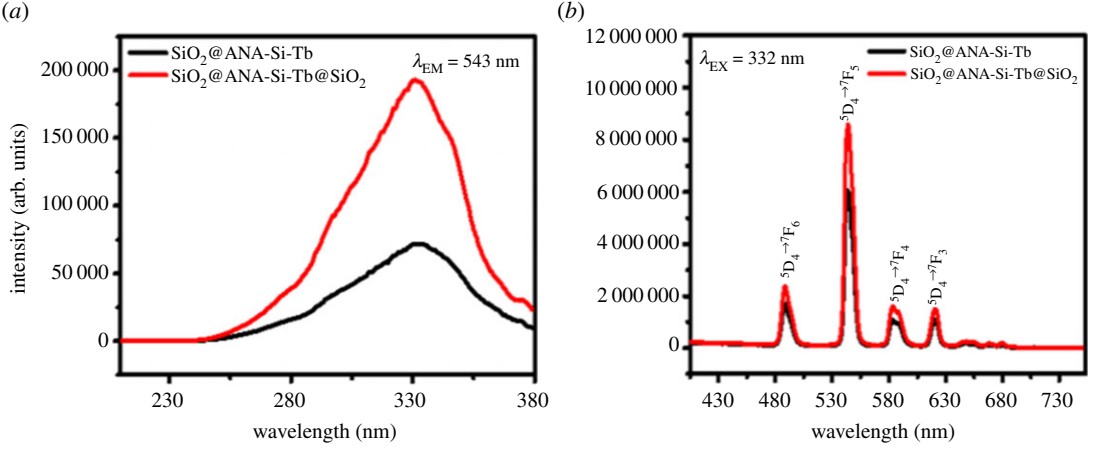

**Figure 9.** Photoluminescence excitation spectra (*a*) and emission spectra (*b*) of SiO$_2$@ANA-Si-Tb (black line) and SiO$_2$@ANA-Si-Tb@SiO$_2$ (red line).

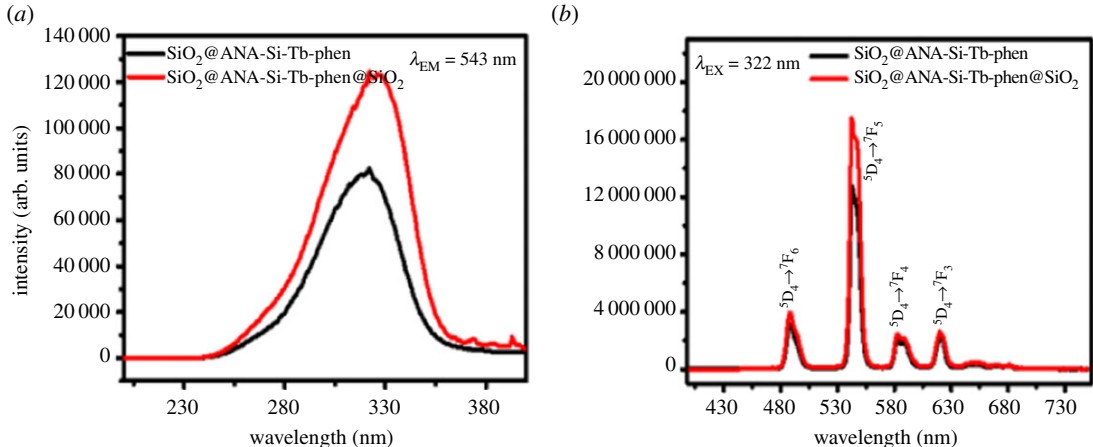

**Figure 10.** Photoluminescence excitation spectra (*a*) and emission spectra (*b*) of SiO$_2$@ANA-Si-Tb-phen (black line) and SiO$_2$@ANA-Si-Tb-phen@SiO$_2$ (red line).

After the incorporation of a second ligand (phen or TTA) and SiO$_2$ shell, the growth of terbium organic complex and SiO$_2$ shell onto the core–shell and core–shell–shell nanostructured composites had been characterized by XRD. Electronic supplementary material, figure S2(A)a–d shows the XRD patterns of SiO$_2$, SiO$_2$@ANA-Si, SiO$_2$@ANA-Si-Tb-phen and SiO$_2$@ANA-Si-Tb-phen@SiO$_2$. Compared with the diffraction pattern of SiO$_2$ core (electronic supplementary material, figure S2(A)a), the SiO$_2$@ANA-Si-Tb-phen (electronic supplementary material, figure S2(A)c) showed several diffraction peaks at about 8.1°, 9.4°, 12.5°, 19.0°, 21.6°, 23.9° and 27.2°, implying that the Tb-(ANA-Si)-phen-ClO$_4$ complexes coated onto SiO$_2$. The diffraction peaks of SiO$_2$@ANA-Si-Tb-phen@SiO$_2$ (electronic supplementary material, figure S2(A)d) displayed the disappearance of several diffraction peaks. However, it was well matched with SiO$_2$ core, indicating that an amorphous SiO$_2$ was coated around the core–shell nanostructured composite. Electronic supplementary material, figure S2(B)a–d shows the XRD patterns of SiO$_2$, SiO$_2$@ANA-Si, SiO$_2$@ANA-Si-Tb-TTA and SiO$_2$@ANA-Si-Tb-TTA@SiO$_2$. It was very similar to that of electronic supplementary material, figure S2(A), which also illustrated that an amorphous SiO$_2$ was coated onto the surface of SiO$_2$@ANA-Si-Tb-TTA.

## 3.5. Photoluminescence properties

To evaluate the effect of surface modification of SiO$_2$ on optical absorption properties, photoluminescence spectra were performed to get more information about optical behaviour of the Tb(III) core–shell and core–shell–shell nanostructured composites. As shown in figures 9–11, the room temperature photoluminescence excitation and emission spectra of SiO$_2$@ANA-Si-Tb, SiO$_2$@ANA-Si-Tb-phen, SiO$_2$@ANA-Si-Tb-TTA and

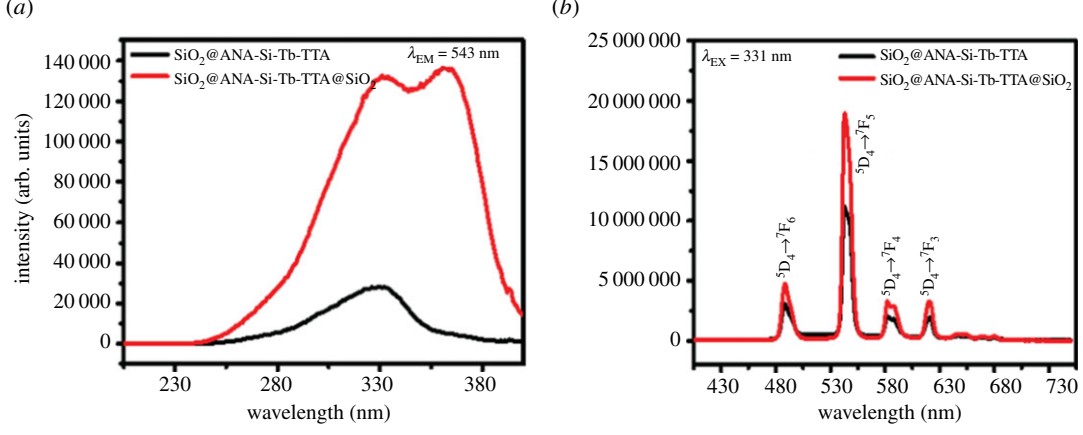

**Figure 11.** Photoluminescence excitation spectra (*a*) and emission spectra (*b*) of SiO$_2$@ANA-Si-Tb-TTA (black line) and SiO$_2$@ANA-Si-Tb-TTA@SiO$_2$ (red line).

SiO$_2$@ANA-Si-Tb@SiO$_2$, SiO$_2$@ANA-Si-Tb-phen@SiO$_2$, SiO$_2$@ANA-Si-Tb-TTA@SiO$_2$ were investigated in solid powder state with the slit width of 0.7 nm. Electronic supplementary material, table S1 shows the photoluminescence emission spectra data of Tb(III) core–shell and core–shell–shell nanostructured composites.

The excitation spectra monitored at 543 nm was investigated in figures 9–11*a*. It can be clearly observed that all excitation spectra exhibited a very broadband at the region of 200–400 nm with the maximum excitation wavelength centred at about 332, 322 and 331 nm, respectively. The broad absorption bands correspond to the 4f$^8$–4f$^7$5d transition of Tb(III). Meanwhile, the corresponding emission spectra were also measured in figures 9–11*b*. These sharp emission peaks at about 488, 543, 583 and 621 nm were related to $^5D_4 \rightarrow {}^7F_6$, $^5D_4 \rightarrow {}^7F_5$, $^5D_4 \rightarrow {}^7F_4$ and $^5D_4 \rightarrow {}^7F_3$ transitions of terbium ions, which the $^5D_4 \rightarrow {}^7F_5$ transition at about 543 nm was the strongest [49–52]. As shown in electronic supplementary material, table S1, the core–shell and core–shell–shell nanostructured composites both exhibited much stronger luminescent properties. The strongest emission intensity of SiO$_2$@ANA-Si-Tb@SiO$_2$, SiO$_2$@ANA-Si-Tb-phen@SiO$_2$ and SiO$_2$@ANA-Si-Tb-TTA@SiO$_2$ was 8 584 812 arb. units, 17 476 444 arb. units and 18 939 780 arb. units, respectively. The emission intensity of SiO$_2$@ANA-Si-Tb@SiO$_2$, SiO$_2$@ANA-Si-Tb-phen@SiO$_2$ and SiO$_2$@ANA-Si-Tb-TTA@SiO$_2$ was about 1.42, 1.38 and 1.70 times, respectively, higher, compared with the emission intensity of the corresponding terbium core–shell nanostructured composites. This result was consistent with fluorescence quantum yield measurements. The absolute quantum yields of SiO$_2$@ANA-Si-Tb, SiO$_2$@ANA-Si-Tb-phen and SiO$_2$@ANA-Si-Tb-TTA were 13.75, 18.22 and 22.32%, respectively, while those of the SiO$_2$@ANA-Si-Tb@SiO$_2$, SiO$_2$@ANA-Si-Tb-phen@SiO$_2$ and SiO$_2$@ANA-Si-Tb-TTA@SiO$_2$ were 23.18, 25.57 and 31.95%, respectively. It was implied that the terbium core–shell nanostructured composites were coated with an amorphous silica shell, which the high vibration energies loss of the ligand molecules (ANA-Si, phen, TTA) or other quenching sites located at the surface of the terbium core–shell nanostructured composites could largely reduce. As a result, the emission efficiency of core–shell–shell nanostructured composite was significantly enhanced. Furthermore, the introduction of phen and TTA can also sensitize the luminescence of terbium ions by 'antenna effect' and largely improve the luminescent properties of the core–shell nanostructured composites.

Further to investigate the photoluminescence stability of the core–shell–shell nanostructured composites in aqueous solution, SiO$_2$@ANA-Si-Tb@SiO$_2$, SiO$_2$@ANA-Si-Tb-phen@SiO$_2$ and SiO$_2$@ANA-Si-Tb-TTA@SiO$_2$ were dissolved in deionized water at a concentration of 0.1 g l$^{-1}$. The luminescence properties were observed after placing for 0, 16 and 40 h at room temperature with the slit width of 2.0 nm. The aqueous solution was sonicated for half an hour before each measurement to achieve a homogeneous suspension. Figure 12*a–c* displays the room temperature photoluminescence emission spectra of the SiO$_2$@ANA-Si-Tb@SiO$_2$, SiO$_2$@ANA-Si-Tb-phen@SiO$_2$ and SiO$_2$@ANA-Si-Tb-TTA@SiO$_2$ after being placed in aqueous solution for 0, 16 and 40 h. It was clearly demonstrated that the emission spectra showed four typical peaks at about 488, 543, 583, 621 nm. Three novel core–shell–shell composites still maintained excellent luminescent properties, and the core–shell–shell structure was not damaged even after 40 h. This result could be attributed to the action of the core–shell nanostructured composites surface covered an amorphous silica shell that protected the

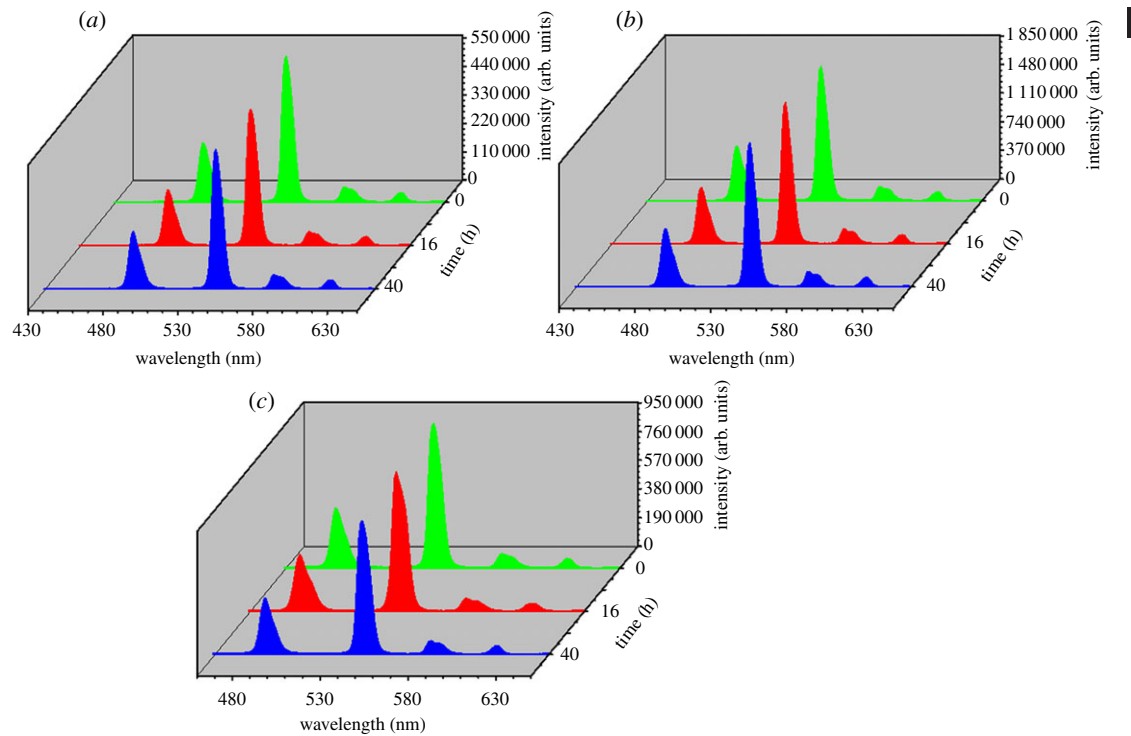

**Figure 12.** Photoluminescence emission spectra of SiO$_2$@ANA-Si-Tb@SiO$_2$ (*a*), SiO$_2$@ANA-Si-Tb-phen@SiO$_2$ (*b*) and SiO$_2$@ANA-Si-Tb-TTA@SiO$_2$ (*c*) after placement for 0, 16 and 40 h in aqueous solution.

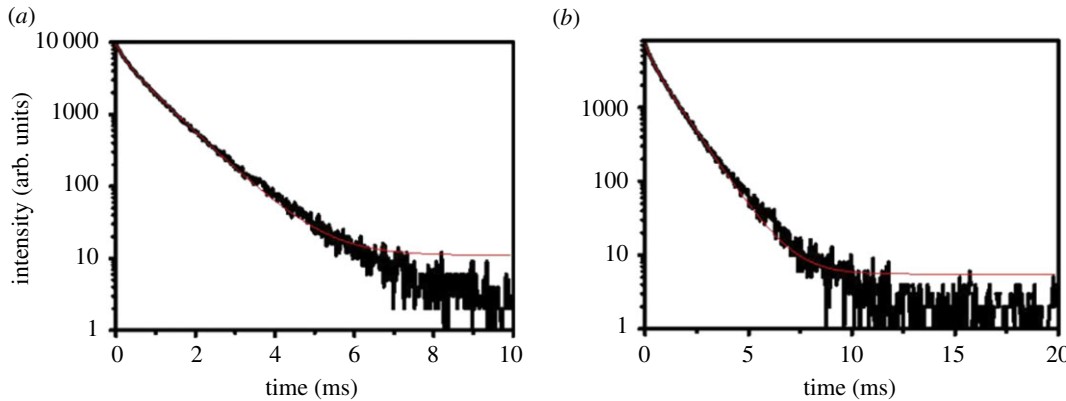

**Figure 13.** Lifetime curve of SiO$_2$@ANA-Si-Tb-phen (*a*) and SiO$_2$@ANA-Si-Tb-phen@SiO$_2$ (*b*).

luminescence centre from the surrounding environment. Therefore, it is worth pointing out that the as-prepared terbium core–shell–shell nanostructured composites exhibited excellent luminescence and high luminescent stability, which could be used as bio-imaging and optical probe.

## 3.6. The photoluminescence lifetime

In order to see the details of photoluminescence property of terbium core–shell and core–shell–shell nanostructured composites, the photoluminescence lifetime curves of SiO$_2$@ANA-Si-Tb-phen and SiO$_2$@ANA-Si-Tb-phen@SiO$_2$ are recorded as shown in figure 13. Simultaneously, the photoluminescence lifetime curves of SiO$_2$@ANA-Si-Tb, SiO$_2$@ANA-Si-Tb@SiO$_2$, SiO$_2$@ANA-Si-Tb-TTA and SiO$_2$@ANA-Si-Tb-TTA@SiO$_2$ are shown in electronic supplementary material, figure S3.

It can be seen clearly that the decay curve is well fitted into a biexponential function (3.1). The lifetime values of the excited state terbium ion ($^5D_4$) could be calculated using the following equation (3.2), where

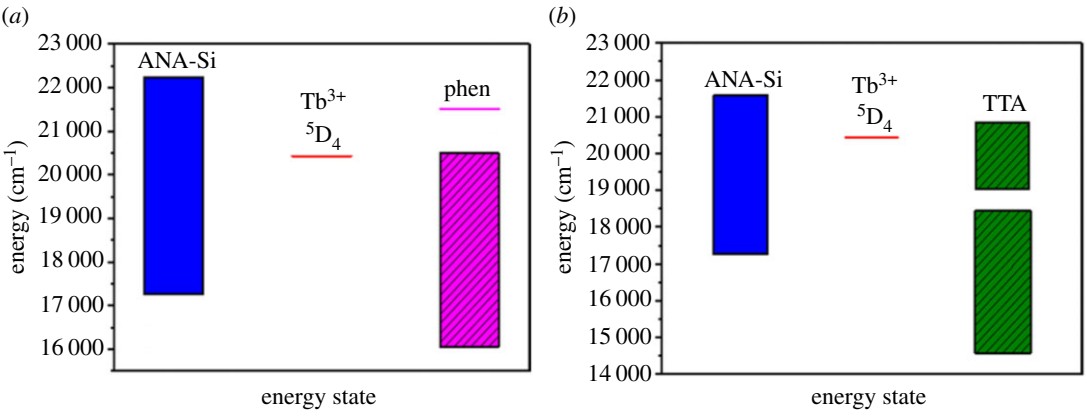

**Figure 14.** Triplet state of ANA-Si, phen and the excited state of Tb(III) (*a*), triplet state of ANA-Si, TTA and the excited state of Tb(III) (*b*).

$\tau_1$ and $\tau_2$ stand for the slow and fast terms of the luminescent lifetime. $A_1$ and $A_2$ are the corresponding pre-exponential factors. The resulting lifetime data and fitting parameters for these Tb(III) core–shell and core–shell–shell nanostructured composites are shown in electronic supplementary material, table S2.

$$I_{(t)} = I_0 + A_1 \exp\left(-\frac{t_1}{\tau_1}\right) + A_2 \exp\left(-\frac{t_2}{\tau_2}\right) \tag{3.1}$$

and

$$\langle \tau \rangle = \frac{A_1 \tau_1^2 + A_2 \tau_2^2}{A_1 \tau_1 + A_2 \tau_2}. \tag{3.2}$$

Herein, the calculated average lifetime of three terbium core–shell nanostructured composites $SiO_2$@ANA-Si-Tb, $SiO_2$@ANA-Si-Tb-phen and $SiO_2$@ANA-Si-Tb-TTA were 374.03 µs, 763.44 µs and 459.12 µs, respectively. What is more, the average lifetime of three terbium core–shell–shell nanostructured composites $SiO_2$@ANA-Si-Tb@$SiO_2$, $SiO_2$@ANA-Si-Tb-phen@$SiO_2$ and $SiO_2$@ANA-Si-Tb-TTA@$SiO_2$ were 375.24 µs, 976.40 µs and 628.10 µs, respectively. The increase in the photoluminescence lifetime for three terbium core–shell–shell nanostructured composites showed that the quenching from outside particles was strongly reduced after the growth of a silica shell around the core–shell nanostructured composites. Three terbium core–shell–shell nanostructured composites have the advantage of long lifetimes under UV excitation. Therefore, they can be used as fluorescent labels to examine the bio-molecules.

## 3.7. Low-temperature phosphorescence analysis

To verify the intramolecular energy transfer from the triplet state of the organic ligand to the resonance level of rare earth ion, the phosphorescence spectrum of the first ligand ANA-Si and second ligand phen and TTA was measured under irradiation of 369, 372 and 395 nm UV lamp, with slit widths of 5, 10 and 10 nm in solid powder state at 77 K (electronic supplementary material, figure S4). There was a broad emission band in the phosphorescence spectrum of ANA-Si, just as shown in electronic supplementary material, figure S4(a). The range of the triplet state energy level was calculated using the emission peak width at the half peak height of 464 and 578 nm. The triplet state energy level of ANA-Si was 21 552–17 301 cm$^{-1}$, which it was higher than $^5D_4$ of Tb(III) ions (20 430 cm$^{-1}$) [53] (figure 14*a*). The first ligand ANA-Si could effectively transfer the energy absorbed in the ultraviolet region to the central terbium ions by non-radiative transition mode, sensitizing the luminescence of terbium ions. Therefore, the $SiO_2$@ANA-Si-Tb core–shell nanostructured composites had excellent luminescence.

The phosphorescence spectrum of phen is shown in electronic supplementary material, figure S4(b), and there were two phosphorescence emission bands. The first band ranged from 484 to 700 nm, which was defined as $T_1$. The triplet state energy level of $T_1$ was 19 920–16 077 cm$^{-1}$. The second band was at 465 nm, which was defined as $T_2$. The triplet state energy level of $T_2$ was 21 505 cm$^{-1}$. Furthermore, the

phosphorescence spectrum of TTA is shown in electronic supplementary material, figure S4(c), and there were two relatively symmetric emission bands between 430 and 700 nm. The first band ranged from 525 to 700 nm was defined as $T_1$. The second band ranged from 480 to 525 nm was defined as $T_2$. The triplet state energy level of $T_1$ was 18 450–14 599 $cm^{-1}$, and the triplet state energy level of $T_2$ was 20 833–19 048 $cm^{-1}$. These results suggested that the triplet state energy level of the second ligand (phen and TTA) and the excited state energy of terbium ions matched very well (figure 14$a,b$). The second ligand could also effectively transfer the energy absorbed in the ultraviolet region to the central terbium ions and sensitize the luminescence of terbium ions. Therefore, the photoluminescence emission intensities of SiO$_2$@ANA-Si-Tb-phen and SiO$_2$@ANA-Si-Tb-TTA were stronger than those of the SiO$_2$@ANA-Si-Tb core–shell nanostructured composite.

# 4. Conclusion

Three novel SiO$_2$@ANA-Si-Tb, SiO$_2$@ANA-Si-Tb-phen and SiO$_2$@ANA-Si-Tb-TTA core–shell nanostructured composites were successfully synthesized with organosilane ANA-Si acting as a 'bifunctional bridge molecular'. Moreover, silica-modified SiO$_2$@ANA-Si-Tb@SiO$_2$, SiO$_2$@ANA-Si-Tb-phen@SiO$_2$ and SiO$_2$@ANA-Si-Tb-TTA@SiO$_2$ core–shell–shell nanostructured composites were successfully synthesized. The possible mechanism for the formation of the core–shell and core–shell–shell nanostructured composites had been investigated in detail. The core–shell and core–shell–shell nanostructured composites both exhibited a much stronger luminescent properties in the solid state. The photoluminescence emission intensity of SiO$_2$@ANA-Si-Tb@SiO$_2$, SiO$_2$@ANA-Si-Tb-phen@SiO$_2$ and SiO$_2$@ANA-Si-Tb-TTA@SiO$_2$ was about 1.42, 1.38 and 1.70 times, respectively, compared with that of the homologous terbium core–shell nanostructured composites. The SiO$_2$ layers were coated on the surface of core–shell nanostructured composites, a better luminescence stability and stronger luminescent properties for the core–shell–shell nanostructured composites were found in the aqueous medium. It may be beneficial for the application in the biomedical field.

Data accessibility. All the experimental data are included in the manuscript. The data are available in the Dryad Digital Repository at: https://doi.org/10.5061/dryad.qm8b161 [54].
Authors' contributions. L.F. carried out the molecular laboratory work, participated in data analysis, carried out sequence alignments, participated in the design of the study and drafted the manuscript; W.L. and Y.Z. carried out the statistical analyses; J.B., Y.L. and Y.M. collected field data; K.Y., Y.Q. and A.W. conceived of the study, designed the study, coordinated the study and helped draft the manuscript. All the authors gave their final approval for publication.
Competing interests. We declare that we have no competing interests.
Funding. This work was supported by the Major projects of Natural Science Foundations of Inner Mongolia Science Foundation (no. 2015ZD01) and the Natural Science Foundations of Inner Mongolia Science Foundation (no. 2015MS0502).
Acknowledgements. All the people who contributed to the study are listed as co-authors.

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
