## [Reviewer comments · Royal Society Open Science]

Review History

RSOS-190182.R0 (Original submission)

Review form: Reviewer 1

Is the manuscript scientifically sound in its present form?

No

Are the interpretations and conclusions justified by the results?

Yes

Is the language acceptable?

No

Is it clear how to access all supporting data?

Yes

Do you have any ethical concerns with this paper?

No

Have you any concerns about statistical analyses in this paper?

No

Recommendation?

Major revision is needed (please make suggestions in comments)

Comments to the Author(s)

The authors describe the preparation of the core-shell and core-shell-shell nanosystems composed of a silica core and Tb complexes on the surface. A second silica shell was introduced to stabilize the luminescent entities and improve the properties of photoluminescence. The novelty compared to the previous systems reported in the literature is the incorporation of the second inorganic layer. This strategy is interesting and its effectiveness has been demonstrated by the authors. This topic can be interesting for a wide range of readers and the manuscript is suitable for publication after some suggested changes reported in the attached file (Appendix A).

Review form: Reviewer 2

Is the manuscript scientifically sound in its present form?

Yes

Are the interpretations and conclusions justified by the results?

Yes

Is the language acceptable?

Yes

Is it clear how to access all supporting data?

Yes

Do you have any ethical concerns with this paper?

No

Have you any concerns about statistical analyses in this paper?

No

Recommendation?

Accept with minor revision (please list in comments)

Comments to the Author(s)

The manuscript entitled "Synthesis and photoluminescence properties of Silica-modified SiO₂@ANA-Si-Tb@SiO₂, SiO₂@ANA-Si-Tb-L@SiO₂ core-shell-shell nanostructured composites" reports the synthesis and characterization of new core-shell-shell nanostructures, which show better luminescence stability and stronger luminescent properties than their homologous terbium core-shell nanostructured composites. This work is continuation of previous papers of this research group [29,44].

In my opinion, the major weakness is the proposed potential biomedical application, as bio-imaging and optical probe for medical diagnosis. Both the large particle size of the core-shell-shell nanostructures (around 200 nm) and the UV irradiation are two important inconveniences for its biomedical use. For instance, the particle size affects directly to its blood circulation time

specially for inorganic nanoparticles [e.g. Nanoparticles for biomedical imaging, *Expert Opin. Drug Deliv.* 2009, 6, 1175–1194] while the UV irradiation can produce cell damage and DNA mutations [e.g. The Mechanisms of UV Mutagenesis. *J. Radiat. Res.* 2011, 52, 115–125]. Therefore, I would recommend to the authors the proposition of other type of applications, or they should include several references, demonstrating that similar nanostructures (i.e. both with similar inorganic-particle size and excitation by UV radiation) were used in photonic-based biomedical sciences.

Other minor comments:

- (1) The authors use too many abbreviations both in the title (e.g. ANA, Tb and L) and in the abstract (phen, TTA, TEOS, CTAB, etc.). They should avoid the use of abbreviations in the title or define them previously in the abstract.
- (2) Figure 2 should include the RDX spectra of SiO₂ and SiO₂@ANA-Si.
- (3) EDX plots included in Figure 3 and 4 have poor quality. Authors should improve these EDX graphical representations.
- (4) Figure 10 shows photoluminescence emission spectra of three different core-shell-shell nanostructures after different incubation times in aqueous solution. In all cases, the spectra are overlapped, and it is impossible to appreciate any difference between different times.

Decision letter (RSOS-190182.R0)

26-Mar-2019

Dear Professor Li:

Title: Synthesis and Photoluminescence Properties of Silica-modified SiO₂@ANA-Si-Tb@SiO₂, SiO₂@ANA-Si-Tb-L@SiO₂ Core-Shell-Shell Nanostructured Composites
Manuscript ID: RSOS-190182

The editor assigned to your manuscript has now received comments from reviewers. We would like you to revise your paper in accordance with the referee and Subject Editor suggestions which can be found below (not including confidential reports to the Editor). Please note this decision does not guarantee eventual acceptance.

Please submit your revised paper before 18-Apr-2019. Please note that the revision deadline will expire at 00.00am on this date. If we do not hear from you within this time then it will be assumed that the paper has been withdrawn. In exceptional circumstances, extensions may be possible if agreed with the Editorial Office in advance. We do not allow multiple rounds of revision so we urge you to make every effort to fully address all of the comments at this stage. If deemed necessary by the Editors, your manuscript will be sent back to one or more of the original reviewers for assessment. If the original reviewers are not available we may invite new reviewers.

When submitting your revised manuscript, you must respond to the comments made by the

referees and upload a file "Response to Referees" in "Section 6 - File Upload". Please use this to document how you have responded to the comments, and the adjustments you have made. In order to expedite the processing of the revised manuscript, please be as specific as possible in your response.

Please also include the following statements alongside the other end statements. As we cannot publish your manuscript without these end statements included, if you feel that a given heading is not relevant to your paper, please nevertheless include the heading and explicitly state that it is not relevant to your work.

- Ethics statement

Please clarify whether you received ethical approval from a local ethics committee to carry out your study. If so please include details of this, including the name of the committee that gave consent in a Research Ethics section after your main text. Please also clarify whether you received informed consent for the participants to participate in the study and state this in your Research Ethics section.

OR

Please clarify whether you obtained the necessary licences and approvals from your institutional animal ethics committee before conducting your research. Please provide details of these licences and approvals in an Animal Ethics section after your main text.

OR

Please clarify whether you obtained the appropriate permissions and licences to conduct the fieldwork detailed in your study. Please provide details of these in your methods section.

- Data accessibility

It is a condition of publication that you make available the data and research materials supporting the results in the article. Datasets should be deposited in an appropriate publicly available repository and details of the associated accession number, link or DOI to the datasets must be included in the Data Accessibility section of the article (<http://royalsocietypublishing.org/instructions-authors#question17>). Reference(s) to datasets should also be included in the reference list of the article with DOIs (where available).

Please include a Data Availability section after your main text stating where supporting data are available from, or where they will be made available should your article be accepted for publication.

If you wish to submit your supporting data or code to Dryad (<http://datadryad.org/>), or modify your current submission to dryad, please use the following link:
<http://datadryad.org/submit?journalID=RSOS&manu=RSOS-190182>

On behalf of the Subject Editor Professor Anthony Stace and the Associate Editor Professor Claire Carmalt.

RSC Associate Editor:
Comments to the Author:
(There are no comments.)

RSC Subject Editor:
Comments to the Author:
(There are no comments.)

Reviewers' Comments to Author:
Reviewer: 1

Comments to the Author(s)

The authors describe the preparation of the core-shell and core-shell-shell nanosystems composed of a silica core and Tb complexes on the surface. A second silica shell was introduced to stabilize the luminescent entities and improve the properties of photoluminescence. The novelty compared to the previous systems reported in the literature is the incorporation of the second inorganic layer. This strategy is interesting and its effectiveness has been demonstrated by the authors. This topic can be interesting for a wide range of readers and the manuscript is suitable for publication after some suggested changes reported in the attached file.

Reviewer: 2

Comments to the Author(s)

The manuscript entitled "Synthesis and photoluminescence properties of Silica-modified SiO₂@ANA-Si-Tb@SiO₂, SiO₂@ANA-Si-Tb-L@SiO₂ core-shell-shell nanostructured composites" reports the synthesis and characterization of new core-shell-shell nanostructures, which show better luminescence stability and stronger luminescent properties than their homologous terbium core-shell nanostructured composites. This work is continuation of previous papers of this research group [29,44].

In my opinion, the major weakness is the proposed potential biomedical application, as bio-imaging and optical probe for medical diagnosis. Both the large particle size of the core-shell-shell nanostructures (around 200 nm) and the UV irradiation are two important inconveniences for its biomedical use. For instance, the particle size affects directly to its blood circulation time specially for inorganic nanoparticles [e.g. Nanoparticles for biomedical imaging, Expert Opin. Drug Deliv. 2009, 6, 1175–1194] while the UV irradiation can produce cell damage and DNA mutations [e.g. The Mechanisms of UV Mutagenesis. J. Radiat. Res. 2011, 52, 115–125]. Therefore, I would recommend to the authors the proposition of other type of applications, or they should include several references, demonstrating that similar nanostructures (i.e. both with similar inorganic-particle size and excitation by UV radiation) were used in photonic-based biomedical sciences.

Other minor comments:

- (1) The authors use too many abbreviations both in the title (e.g. ANA, Tb and L) and in the abstract (phen, TTA, TEOS, CTAB, etc.). They should avoid the use of abbreviations in the title or define them previously in the abstract.
- (2) Figure 2 should include the RDX spectra of SiO₂ and SiO₂@ANA-Si.

(3) EDX plots included in Figure 3 and 4 have poor quality. Authors should improve these EDX graphical representations.

(4) Figure 10 shows photoluminescence emission spectra of three different core-shell-shell nanostructures after different incubation times in aqueous solution. In all cases, the spectra are overlapped, and it is impossible to appreciate any difference between different times.

Author's Response to Decision Letter for (RSOS-190182.R0)

See Appendix B.

RSOS-190182.R1 (Revision)

Review form: Reviewer 2

Is the manuscript scientifically sound in its present form?

Yes

Are the interpretations and conclusions justified by the results?

Yes

Is the language acceptable?

Yes

Is it clear how to access all supporting data?

Yes

Do you have any ethical concerns with this paper?

No

Have you any concerns about statistical analyses in this paper?

No

Recommendation?

Accept as is

Comments to the Author(s)

The manuscript entitled "Synthesis and Photoluminescence Properties of Silica-modified SiO₂@ANA-Si-Tb@SiO₂, SiO₂@ANA-Si-Tb-L@SiO₂ Core-Shell-Shell Nanostructured Composites" reports the synthesis and characterization of new core-shell-shell nanostructures, which show better luminescence stability and stronger luminescent properties than their homologous terbium core-shell nanostructured composites. This work is continuation of previous papers of this research group [29,44].

I consider that the authors have made enough efforts to answer all reviewer's comments and suggestions, and therefore the manuscript is ready for publication.

Decision letter (RSOS-190182.R1)

10-May-2019

Dear Professor Li:

Title: Synthesis and Photoluminescence Properties of Silica-modified SiO₂@ANA-Si-Tb@SiO₂, SiO₂@ANA-Si-Tb-L@SiO₂ Core-Shell-Shell Nanostructured Composites
Manuscript ID: RSOS-190182.R1

It is a pleasure to accept your manuscript in its current form for publication in Royal Society Open Science. The chemistry content of Royal Society Open Science is published in collaboration with the Royal Society of Chemistry.

RSC Associate Editor:
Comments to the Author:
(There are no comments.)

RSC Subject Editor:
Comments to the Author:
(There are no comments.)

Reviewer(s)' Comments to Author:
Reviewer: 2

Comments to the Author(s)
The manuscript entitled "Synthesis and Photoluminescence Properties of Silica-modified SiO₂@ANA-Si-Tb@SiO₂, SiO₂@ANA-Si-Tb-L@SiO₂ Core-Shell-Shell Nanostructured Composites" reports the synthesis and characterization of new core-shell-shell nanostructures,

which show better luminescence stability and stronger luminescent properties than their homologous terbium core-shell nanostructured composites. This work is continuation of previous papers of this research group [29,44].

I consider that the authors have made enough efforts to answer all reviewer's comments and suggestions, and therefore the manuscript is ready for publication.

Appendix A

The authors describe the preparation of the core-shell and core-shell-shell nanosystems composed of a silica core and Tb complexes on the surface. A second silica shell was introduced to stabilize the luminescent entities and improve the properties of photoluminescence. The novelty compared to the previous systems reported in the literature is the incorporation of the second inorganic layer. This strategy is interesting and its effectiveness has been demonstrated by the authors. This topic can be interesting for a wide range of readers and the manuscript is suitable for publication after some suggested changes here.

Major comments:

1) The authors exploit the EDX analysis to determine the load of Tb on the surface. This approach is suitable for a preliminary knowledge of the chemical composition. However, a more precise quantification of metal ions should be performed by ICP-MS or ICP-OES analysis. These tests must be performed for samples and data reported in the revised version.

2) IR section: Spectra should be normalized to compare the bands absorbance for all samples. Otherwise, it is difficult to confirm that the SiO₂ shell was successfully created on the surface of the core-shell materials. Furthermore, in paragraph 3.3.2, the authors should insert the relative figure, currently reported in the ESI, in order to facilitate the comprehension of the peaks attribution.

3) The comments about the XRD data should be revised and probably smoothed because also the pattern of SiO₂@ANA-Si-Tb shows a weak band at 7-8° 2theta. The differences in the pattern reported in the figure 6 are mainly related to the intensity ratios that change in the samples.

4) Photoluminescence section:

- the excitation bands at 332, 322 and 331 nm require a precise attribution.

- Authors should describe the methodology used to determine the quantum yields of solid state samples.

- Luminescence stability is defined by photobleaching tests in which the luminescence intensity is followed as a function of time under continuous light irradiation. This test should be performed in order to demonstrate that the luminescent stability of the core-shell-shell sample is better than the core-shell materials.

Minor comments:

The English should be deeply revised. Some grammatical errors are for instance indicated:

1) in the abstract, line 45: "all" should be removed.

2) page 2, line 28: please remove "improved" before "stability"

3) page 2, line 32: "silicon" should be replaced by "silica"

4) page 3, lines 19 and 45: please replace luminous with luminescent

5) page 3, lines 47-51: the sentence is not clear.

6) page 7, line 18: please replace micrograph with micrographs

7) page 7, lines 36-40: the sentence is not clear.

.....

Appendix B

Dear Editor,

I quite appreciate your favorite consideration and the reviewer's insightful comments. Now I have revised the RSOS-190182 exactly according to the reviewer's comments. Those comments are all valuable and very helpful for revising and improving our paper, as well as the important guiding significance to our researches. We have studied comments carefully and have made correction which we hope meet with approval. Revised portions are marked in red in the paper. The main corrections in the paper and the responds to the reviewer's comments are as follows.

Comments to Reviewer 1:

We want to begin by thanking Reviewer 1 for writing that "The authors describe the preparation of the core-shell and core-shell-shell nanosystems composed of a silica core and Tb complexes on the surface. A second silica shell was introduced to stabilize the luminescent entities and improve the properties of photoluminescence. The novelty compared to the previous systems reported in the literature is the incorporation of the second inorganic layer. This strategy is interesting and its effectiveness has been demonstrated by the authors. This topic can be interesting for a wide range of readers and the manuscript is suitable for publication after some suggested changes here". We also appreciated the constructive criticism and suggestion. We addressed all the points raised by the reviewer, as summarized below.

Point 1: The authors exploit the EDX analysis to determine the load of Tb on the surface. This approach is suitable for a preliminary knowledge of the chemical composition. However, a more precise quantification of metal ions should be performed by ICP-MS or ICP-OES analysis. These tests must be performed for samples and data reported in the revised version.

Response 1: We are grateful to the reviewer for pointing out ICP-MS or ICP-OES analysis. We are very sorry for this question. In the core-shell nanostructured composite, the terbium organic complex is acted as the shell layer. Because the complexes containing organic matter could not be measured for ICP-MS or ICP-OES, this test has not been carried out in the revised version. In addition, in order to further study the elemental distribution of the core-shell nanostructured composites, the STEM-EDX elemental mappings have been complemented. It can be seen from figure 4a-d that O and Tb were uniformly distributed in the silica-based spheres-like area.

Furthermore, the element mappings of Si, O and Tb in SiO₂@ANA-Si-Tb composite (figure 4a-d) revealed that O and Tb were uniformly distributed in the silica-based spheres-like area.

Figure 4. Bright-field scanning transmission electron microscopy (STEM) image of a single SiO₂@ANA-Si-Tb (a) and the associated EDX element mappings images (b-d).

Point 2: IR section: Spectra should be normalized to compare the bands absorbance for all samples. Otherwise, it is difficult to confirm that the SiO₂ shell was successfully created on the surface of the core-shell materials. Furthermore, in paragraph 3.3.2, the authors should insert the relative figure, currently reported in the ESI, in order to facilitate the comprehension of the peaks attribution.

Response 2: Thanks for the reviewer's carefully reading and great suggestion. We have revised FT-IR spectra according to the Reviewer's suggestion. The spectra have been normalized to compare the bands absorbance for all samples. Besides, figure 7 have been inserted in the revised manuscript.

Figure 6. FT-IR spectra of ANA-Si (a), SiO₂ (b), SiO₂@ANA-Si (c), SiO₂@ANA-Si-Tb (d) and SiO₂@ANA-Si-Tb@SiO₂ (e).

Figure 7. FT-IR spectra of phen (a), SiO₂@ANA-Si-Tb-phen (b), SiO₂@ANA-Si-Tb-phen@SiO₂ (c), TTA (d), SiO₂@ANA-Si-Tb-TTA (e) and SiO₂@ANA-Si-Tb-TTA@SiO₂ (f).

Point 3: The comments about the XRD data should be revised and probably smoothed because also the pattern of SiO₂@ANA-Si-Tb shows a weak band at 7-8° 2theta. The differences in the pattern reported in the figure 6 are mainly related to the intensity ratios that change in the samples.

Response 3: Thanks for the reviewer's suggestion very much. We have revised this part according to the Reviewer's suggestion as follows.

After surface modification with organosilane (ANA-Si), as shown in figure 8b, no diffraction peak was observed except for the broad bands, which were the characteristic peak for amorphous SiO₂. The XRD patterns of SiO₂@ANA-Si-Tb (figure 8c), the intensity of the diffraction pattern appeared at $2\theta = 7-8^\circ$ became weaker. Careful view showed that there were some weak peaks centered at 6.4° , 7.9° , 9.1° , 10.2° and 11.1° , also providing evidence for the Tb-(ANA-Si)-ClO₄ complexes coated onto the surface of SiO₂ core. As illustrated in figure 8d, the two broad bands of SiO₂@ANA-Si-Tb@SiO₂ core-shell-shell nanostructured composite appeared at $2\theta = 7-8^\circ$ and 23° were in coincidence with the stander XRD pattern for SiO₂ core (figure 8a), which further illustrated that an amorphous SiO₂ was coated onto the surface of SiO₂@ANA-Si-Tb.

Figure 8. XRD pattern of SiO₂ (a), SiO₂@ANA-Si (b), SiO₂@ANA-Si-Tb (c) and SiO₂@ANA-Si-Tb@SiO₂ (d).

Point 4: The excitation bands at 332, 322 and 331 nm require a precise attribution.

Response 4: We are very grateful for the reviewer's suggestion. Further to investigate the attribution of excitation bands at 332, 322 and 331 nm. The broad absorption bands correspond to the $4f^8-4f^75d$ transition of Tb(III).

Point 5: Authors should describe the methodology used to determine the quantum yields of solid state samples.

Response 5: We are very grateful for the reviewer's suggestion. The detailed description of the test methodology has been added in the revised manuscript as follows.

2.8. Test methodology for the quantum yields

The absolute fluorescence quantum yield of the solid powder sample was measured by an integrating sphere using the photoluminescence spectrometer (FL; Edinburgh S980, UK) at room temperature. First, the excitation and emission spectra of the sample should be acquired in order to identify the best excitation wavelength and

emission range. Subsequently, the integrating sphere was put in the sample chamber. In powder samples, the blank sample measurements were made by replacing the PTFE powder vessel with the BENFLEC scattering platform. An emission spectrum covering the excitation light and the full emission spectrum was recorded. This measurement should be done with 0.1 nm step size and 0.3 second integration time, 1-5 repeats (depending on the accuracy required). Monitor the counts per second on the emission detector until there was sufficient intensity on the detector (500000-1 million cps). Then the emission spectrum of the solid powder samples covering the excitation light and the full emission spectrum were recorded. This measurement should be done with the same settings as before. Finally, the quantum yield function was used to select the spectra and define the calculation region for scatter and emission. After this was completed, the quantum yield was obtained.

Point 6: Luminescence stability is defined by photobleaching tests in which the luminescence intensity is followed as a function of time under continuous light irradiation. This test should be performed in order to demonstrate that the luminescent stability of the core-shell-shell sample is better than the core-shell materials.

Response 6: Thanks for the helpful comments and suggestions. We are very sorry for this question. Owing to the limitation of experimental conditions, the photobleaching test had not been measured. Meanwhile, to further investigate the photoluminescence stability of the core-shell-shell nanostructured composites in aqueous solution, the $\text{SiO}_2@ANA\text{-Si-Tb-phen}@SiO_2$ composite was dissolved in deionized water at a concentration of 0.1 g L^{-1} and under the continuous UV light irradiation at 365 nm. The luminescence property was observed after irradiated 0 h, 16 h, 40 h at room temperature using the photoluminescence spectrometer (FL; Edinburgh S980, UK) with the slit width of 2.0 nm. With the time increasing from 0 h to 40 h, the decrease degree of emission intensity was smaller. The core-shell-shell composite also exhibited better luminescence stability under the continuous UV light irradiation. This result was consistent with the photoluminescence stability of the core-shell-shell nanostructured composite in figure 12 under intermittently UV light irradiation. Therefore, the introduction of an amorphous silica shell not only can suppress OH groups quenching effect from the surrounding environment, but it can also protect the internal structure of the core-shell composites.

Photoluminescence emission spectra of SiO₂@ANA-Si-Tb-phen@SiO₂ after placement 0 h, 16 h and 40 h in aqueous solution and under the continuous UV light irradiation at 365 nm.

Point 7: Minor comments: The English should be deeply revised. Some grammatical errors are for instance indicated. In the abstract, line 45: “all” should be removed.

Response 7: Thanks for the helpful comments and suggestions. Throughout the article, we have carefully examined and made reasonable corrections and grammatical corrections. Moreover, in the abstract, line 45: “all” has been removed.

Point 8: page 2, line 28: please remove “improved” before “stability”.

Response 8: Thanks for the helpful suggestion. We have removed the “improved” before “stability”.

Point 9: page 2, line 32: “silicon” should be replaced by “silica”.

Response 9: We are very grateful for the reviewer’s crucial comment. We have replaced the statements of “silicon” by “silica”.

Point 10: page 3, lines 19 and 45: please replace luminous with luminescent.

Response 10: We are very grateful for the reviewer’s suggestion. We have replaced the statements of “luminous” by “luminescent” in lines 19 and 45.

Point 11: page 3, lines 47-51: the sentence is not clear.

Response 11: Thanks for the reviewer’s comment. This sentence has been revised for more clearly description as follows.

In addition, such silica-modified core-shell-shell nanostructured composites simultaneously show excellent properties with regard to non-toxicity and luminescence that plays a significant role in the development of bio-imaging, medical diagnosis and biological labeling [46,47].

Point 12: page 7, line 18: please replace micrograph with micrographs.

Response 12: We are very grateful for the reviewer’s excellent suggestion. We have

revised the statements of “micrograph” by “micrographs”.

Point 13: page 7, lines 36-40: the sentence is not clear.

Response 13: Thanks for the suggestion. We have revised this sentence for more clearly description as follows.

In order to study the elemental composition of the core-shell nanostructured composites, the as-prepared SiO₂@ANA-Si-Tb, SiO₂@ANA-Si-Tb-phen and SiO₂@ANA-Si-Tb-TTA microspheres were subsequently analyzed by EDX spectrometer in figure 3d and figure S1a. The presence of Si, O, N, Cl and Tb atoms in the EDX spectra suggested the formation of the SiO₂@ANA-Si-Tb, SiO₂@ANA-Si-Tb-phen and SiO₂@ANA-Si-Tb-TTA core-shell nanostructured composites.

Comments to Reviewer 2:

We should like to express my appreciation to reviewer 2 for suggesting how to improve our paper. We have made extensive modification on the original manuscript. The main corrections in the paper and the responds to the reviewer’s comments are as follows.

Point 1: In my opinion, the major weakness is the proposed potential biomedical application, as bio-imaging and optical probe for medical diagnosis. Both the large particle size of the core-shell-shell nanostructures (around 200 nm) and the UV irradiation are two important inconveniences for its biomedical use. For instance, the particle size affects directly to its blood circulation time specially for inorganic nanoparticles [e.g. Nanoparticles for biomedical imaging, Expert Opin. Drug Deliv. 2009, 6, 1175–1194] while the UV irradiation can produce cell damage and DNA mutations [e.g. The Mechanisms of UV Mutagenesis. J. Radiat. Res. 2011, 52, 115–125]. Therefore, I would recommend to the authors the proposition of other type of applications, or they should include several references, demonstrating that similar nanostructures (i.e. both with similar inorganic-particle size and excitation by UV radiation) were used in photonic-based biomedical sciences.

Response 1: We are very grateful for the reviewer’s crucial comment. For biological applications, the prepared core-shell-shell nanostructured composites should be luminescent stability and water-solubility. The silica shell can not only prevent rare earth core-shell nanostructured composites from quenching of the external environment, but also improve the solubility and luminescent stability of the core-shell nanostructured composite material. In addition, these core-shell-shell composites with surface-attached abundant Si-OH groups are easily available for

conjugation with biomolecules. Therefore, they could be used for the realistic testing of the biomedical field. According to the reported literature, there are sufficient researches on core-shell composite materials with SiO₂ as the core and the rare earth complex as cladding layer, which has been used in bio-imaging, medical diagnosis and biological labeling. Here are some references to refer to: (1) J. Chem. Sci., 2016, 128, 1149-1155. (2) J Sol-Gel Sci Technol, 2017, 83, 447-456. (3) Opt. Mater. 2018, 79, 464-469.

Point 2: The authors use too many abbreviations both in the title (e.g. ANA, Tb and L) and in the abstract (phen, TTA, TEOS, CTAB, etc.). They should avoid the use of abbreviations in the title or define them previously in the abstract.

Response 2: Thanks for the suggestion. We have revised this part according to the reviewer's suggestion. The abbreviations both in the title (e.g. ANA, Tb and L) and in the abstract (phen, TTA, TEOS, CTAB, etc.) have been defined previously in the abstract.

Point 3: Figure 2 should include the XRD spectra of SiO₂ and SiO₂@ANA-Si.

Response 3: Thanks for the reviewer's comment. According to the suggestions, we have revised this part. The XRD spectra of SiO₂ and SiO₂@ANA-Si have been inserted in Figure S2.

Figure S2. (A) XRD pattern of SiO₂ (a), SiO₂@ANA-Si (b), SiO₂@ANA-Si-Tb-phen (c), SiO₂@ANA-Si-Tb-phen@SiO₂ (d). (B) XRD patterns of SiO₂ (a), SiO₂@ANA-Si (b), SiO₂@ANA-Si-Tb-TTA (c), SiO₂@ANA-Si-Tb-TTA@SiO₂ (d).

Point 4: EDX plots included in Figure 3 and 4 have poor quality. Authors should improve these EDX graphical representations.

Response 4: We are very grateful for the reviewer's crucial comment. We have revised this part according to the reviewer's suggestion in the EDX spectra. Owing to an amorphous silica layer that was coated onto the surface of SiO₂@ANA-Si-Tb, SiO₂@ANA-Si-Tb-phen and SiO₂@ANA-Si-Tb-TTA core-shell nanostructured composites, eventually causing the reduction of Tb content when compared with the corresponding core-shell nanostructured composites.

Figure 3. TEM images of SiO₂@ANA-Si-Tb (a), SiO₂@ANA-Si-Tb-phen (b), SiO₂@ANA-Si-Tb-TTA (c) and EDX spectrum of SiO₂@ANA-Si-Tb (d).

Figure 5. TEM images of SiO₂@ANA-Si-Tb@SiO₂ (a), SiO₂@ANA-Si-Tb-phen@SiO₂ (b), SiO₂@ANA-Si-Tb-TTA@SiO₂ (c) and EDX spectrum of SiO₂@ANA-Si-Tb@SiO₂ (d).

Figure S1. The EDX spectra of core-shell nanostructured composites (a) and core-shell-shell nanostructured composites (b).

Point 5: Figure 10 shows photoluminescence emission spectra of three different core-shell-shell nanostructures after different incubation times in aqueous solution. In all cases, the spectra are overlapped, and it is impossible to appreciate any difference between different times.

Response 5: We are grateful to the reviewer for pointing out the problem. We have revised this part in Figure 12.

Figure 12. Photoluminescence emission spectra of $\text{SiO}_2@ANA-Si-Tb@SiO_2$ (a), $\text{SiO}_2@ANA-Si-Tb-phen@SiO_2$ (b) and $\text{SiO}_2@ANA-Si-Tb-TTA@SiO_2$ (c) after placement 0 h, 16 h and 40 h in aqueous solution.

Kind regards,

Prof. Wenxian Li

Inner Mongolia Key Laboratory of Chemistry and Physics of Rare Earth Materials,
 School of Chemistry and Chemical Engineering, Inner Mongolia University,
 Hohhot 010021,
 People's Republic of China